# Identification and characterization of a new ensemble of cometary organic molecules

N. Hänni [1] ✉, K. Altwegg[1], M. Combi [2], S. A. Fuselier[3,4], J. De Keyser [5], M. Rubin [1] & S. F. Wampfler [6]

In-situ study of comet 1P/Halley during its 1986 apparition revealed a surprising abundance of organic coma species. It remained unclear, whether or not these species originated from polymeric matter. Now, high-resolution mass-spectrometric data collected at comet 67P/Churyumov-Gerasimenko by ESA's Rosetta mission unveil the chemical structure of complex cometary organics. Here, we identify an ensemble of individual molecules with masses up to 140 Da while demonstrating inconsistency of the data with relevant amounts of polymeric matter. The ensemble has an average composition of $C_1H_{1.56}O_{0.134}N_{0.046}S_{0.017}$, identical to meteoritic soluble organic matter, and includes a plethora of chain-based, cyclic, and aromatic hydrocarbons at an approximate ratio of 6:3:1. Its compositional and structural properties, except for the H/C ratio, resemble those of other Solar System reservoirs of organics —from organic material in the Saturnian ring rain to meteoritic soluble and insoluble organic matter –, which is compatible with a shared prestellar history.

[1] Physics Institute, Space Research & Planetary Sciences, University of Bern, Sidlerstrasse 5, 3012 Bern, Switzerland. [2] Department of Climate and Space Sciences and Engineering, University of Michigan, Ann Arbor, MI, USA. [3] Space Science Directorate, Southwest Research Institute, San Antonio, TX, USA. [4] Department of Physics and Astronomy, The University of Texas at San Antonio, San Antonio, TX, USA. [5] Royal Belgian Institute for Space Aeronomy, BIRA-IASB, Brussels, Belgium. [6] Center for Space and Habitability, University of Bern, Gesellschaftsstrasse 6, 3012 Bern, Switzerland. ✉email: nora.haenni@unibe.ch

Comets are thought to be composed of refractories and ices aggregated in the outer Solar System. However, the recent comet mission Rosetta, targeting the Jupiter-family comet 67P/Churyumov-Gerasimenko (hereafter 67P), delivered evidence that this traditional classification fails to characterize 67P's material appropriately. The spectrum of cometary matter not only turned out to be very rich from a chemical perspective as many surprising molecules were observed in the coma[1], but it also covers a range of volatility: there are super-volatiles (e.g., CO, sublimation temperature $T_{sub} \sim 30$ K), volatiles (e.g., $CO_2$, $NH_3$, or $H_2O$, $T_{sub} \sim 140$ K), semi-volatiles (e.g., ammonium salts with $T_{sub} > 200$ K, e.g., $NH_4Cl$[2,3]), and refractories composed of a mineral and an organic fraction[4].

While the small volatiles, the major constituents of the cometary neutral gas coma, have been studied with high temporal and spatial resolution over a large part of 67P's orbit[5–7], heavier and more complex species are rare in 67P's coma, especially outside 2 au. Only a few are conclusively identified to date, typically correlated with the comet's dust activity[1,8,9]. The current census of neutral cometary coma species ends with the fully saturated linear alkane chain heptane ($C_7H_{16}$, molecular weight (mw) = 100 Da[8]). The refractory dust consists of minerals and most likely macromolecular organics[10], each contributing $\sim 50\%$ by mass[4]. This distribution is consistent with findings from comet 1P/Halley, where so-called CHON grains and mineral grains were found[11].

The structural properties of the complex cometary organics have remained elusive to date. PICCA observations from ESA's Giotto spacecraft, flying by comet 1P/Halley, detected an ensemble of ion signatures corresponding to molecular masses of up to at least 100 Da[12]. These data, however, were interpreted alternatively as to indicate fragmenting polymers, specifically polyoxymethylene (POM)[13–15], or an ensemble of individual molecules[16]. In addition, observations by PUMA on the Vega 1 spacecraft seemed to support the presence of an ensemble of individual organic species associated to decomposing cometary dust[17]. Notably, the resolution of these mass spectrometers was not sufficient to confirm or negate either suggestion. Decomposing polymers were proposed as a possible explanation for distributed density profiles of certain molecules, which did not seem to originate from the nucleus only. In particular, POMs have been suggested as an explanation for distributed formaldehyde and carbon monoxide sources observed during the 1P/Halley fly-by[18]. These species could, in part, be released from the cometary nucleus and from polymer-containing dust grains in the coma. However, later work demonstrated that distributed profiles could also result from short-term variations in the cometary outgassing[19]. Results from analysis of the collected grains from comet 81P/Wild 2 from the Stardust mission showed a relatively high abundance of oxygen and nitrogen and the presence of aromatic moieties[20,21]. However, polymers or complex individual organic molecules could not be directly identified, thus hampering discrimination between the two suggested origins of the high molecular masses at 1P/Halley. High-resolution mass-spectrometric data, collected in the inner coma of comet 67P by the Double Focusing Mass Spectrometer (DFMS), part of the Rosetta Orbiter Spectrometer for Ion and Neutral Analysis (ROSINA[22]) onboard the Rosetta spacecraft, for the first time offer the opportunity to revisit and resolve this long-standing controversy.

In this work, we identify and characterize a new ensemble of complex organic molecules. While this ensemble includes several species identified in a comet for the first time, it excludes relevant amounts of polymeric matter. We use descriptive parameters like the average sum formula, the $sp^2$:$sp^3$ and the $CH_2$:$CH_3$ ratio, as well as the hydrogen deficiency index (HDI) as a basis for a detailed inter-comparison of these cometary organics with other Solar System reservoirs and the Interstellar Medium (ISM). We argue that the observed similarities and differences in this inter-comparison are consistent with a shared pre-solar history of Solar System organics.

## Results and discussion

**Mass-spectrometric measurements at comet 67P.** On 3 August 2015, comet 67P was at 1.249 au from the Sun, just about to reach its perihelion, and the OSIRIS camera registered an exceptionally high dust activity[23]. As ejected cometary particles are being heated up in the Sun to temperatures of up to a few hundreds of Kelvin[24], as compared to roughly 240 K for the nuclear surface[25], we expected enhanced sublimation of also larger organic molecules from the dust particles. The DFMS was measuring the outgassing from both the cometary nucleus as well as the detached particles at a cometocentric distance of around 215 km. While species sublimating from the nucleus were diluted with $r^{-2}$, where r is the cometocentric distance, species sublimating from the dust may expose distributed density profiles with maxima farther away from the nucleus determined by their sublimation temperatures. A particle velocity of $\sim 1$ m/s[26] would give the particle approximately 60 h to travel and heat up before it passes the orbit of Rosetta. Regardless of their origin, neutral coma species were ionized via electron impact (EI) in the DFMS ion source, extracted, and transferred to the detector after passing an electrostatic analyser and a magnetic sector. The molecular ions (M) as well as the charge-sustaining EI ionization (EII) fragments, inherently produced in the EII process, were registered for each mass-per-charge ($m/z$) ratio separately with a mass resolution $m/\Delta m = 3000$ for $m/z = 28$ at 1% of the peak height. Such resolution is sufficient to distinguish, e.g., pure hydrocarbons from heteroatom-bearing species, which we do in the presented analysis. In order to derive the abundance of any species, the ionization cross-section and the fragmentation pattern of all molecules contributing have to be taken into account. As data for the cross sections are mostly unavailable, we give the intensities, corrected for the mass-dependent sensitivity, in arbitrary units (arb. units). Further details on instrumentation, data collection, and data evaluation are reported in the Methods section.

Figure 1 shows a collection of signals of carbon-bearing species registered in the scanned range up to $m/z = 140$. Each signal stems from either a molecular ion of a parent species contained in the cometary coma or from a charge-sustaining EII fragment of such a parent or from a mixture of those. Technical details relevant to the mass-spectrometric analysis of a complex mixture of gaseous species can be found in the Methods section. As signals above $m/z = 140$ are scarce, we expect that the majority of parent species contributing to the presented data have their molecular ion within the scanned mass range. Also, the high-mass signals are almost exclusively caused by carbon-based species. Especially for $m/z > 50$, non-carbon-based species are rare and limited to a few sulphur-bearing species and elemental sulphur. Noble gases Kr and Xe were not detected during this time.

**67P's volatile complex organics budget.** To determine which species contribute to the overall signal intensity, all data must be deconvolved into individual fragmentation patterns. Key to this deconvolution is the sub-group of pure hydrocarbon species, for the following reasons: First, pure hydrocarbons make up the largest group of species, with roughly twice the total intensity of the next most abundant group. That next most abundant group is O-bearing species (CO and $CO_2$ excluded). Thus, pure hydrocarbon fragments of heteroatom-bearing parents have a small impact on the deconvolution process. Second, deconvolving the subset of pure hydrocarbon species separately also comes with

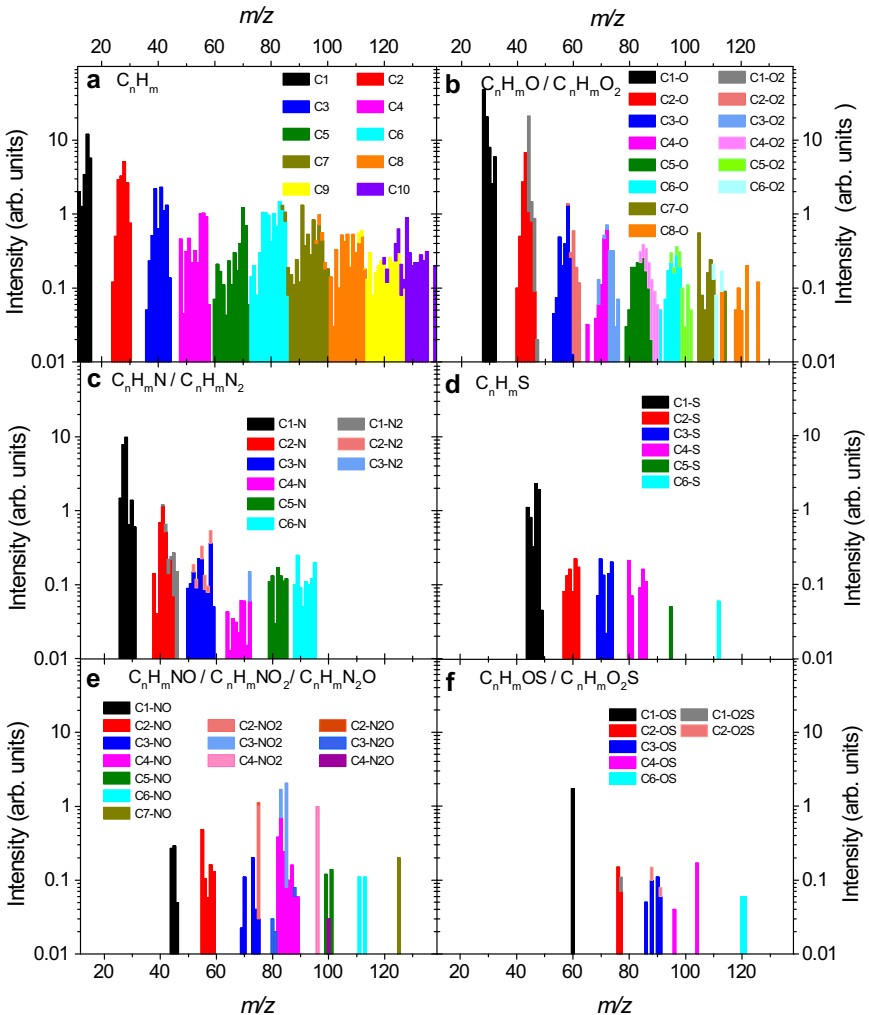

**Fig. 1 C-bearing species identified on 3 August 2015 by the Double Focusing Mass Spectrometer (DFMS).** Signal intensities of parent and fragment species in arbitrary units (arb. units) are plotted as a function of the corresponding integer $m/z$ value. The statistical $1\sigma$ uncertainty of the signals is estimated to be 15–35%, but error bars are omitted for visual clarity. The different panels show the different subgroups of species: **a** pure hydrocarbon species, **b** O-bearing species, **c** N-bearing species, **d** S-bearing species, **e** NO-bearing species, and **f** OS-bearing species. Very rare and low-intensity NS-bearing species are not shown here. $m$ and $n$ indicate variable numbers of C and H atoms, respectively.

minimal uncertainties caused by low-resolution reference data as all fragments must be pure hydrocarbons again, cf. Methods for further details. With a focus on the strongest signals, a few heteroatom-bearing molecules could also be identified - not by deconvolving the full data but based on their characteristic fragmentation patterns. These species are mostly aromatic because aromatic molecules often expose a large M signal and a set of characteristic fragment signals, which does not mean that there are no chain-based heteroatom-bearing molecules at all (for details see the Methods section and Supplementary Table 1). Supplementary Fig. 1 presents an Occam's razor[27] conform deconvolution of the subset of signals associated to pure hydro-carbon species, explaining the largest portion of the observed intensity with the smallest number of contributing molecules (44 in total). Table 1 lists included pure hydrocarbon species along with the heteroatom-bearing candidates identified from the other data.

Among the molecules in Table 1, some are unambiguously identified in a comet for the first time: nonane ($C_9H_{20}$), naphthalene ($C_{10}H_8$), and benzylamine ($C_7H_9N$). Benzoic acid ($C_7H_6O_2$) has been previously reported in Ref. [1] without details and is now clearly confirmed due to the very strong and

characteristic signals (M on $m/z = 122$ and M-OH on $m/z = 105$). As a consequence of the full deconvolution, also two small molecules are confirmed for the first time in comet 67P: ethylene ($C_2H_4$) and propene ($C_3H_6$). Ethylene was reported to be part of comet 1P/Halley's coma[28], while propene has not been detected in any cometary coma previously to the best of our knowledge. Most of the other molecules contained in Table 1 are very likely present but are not confirmed conclusively because reference data of isomers is unavailable or the abundance is low with respect to the error of the signal. In a minority of cases, different isomers cannot be distinguished based on their reference fragmentation patterns. The simultaneous presence of several structural isomers of the same molecule is also possible. Details are explained in the Methods section and in Supplementary Table 1.

Despite the above caveats, our analysis demonstrates that three groups of pure hydrocarbon molecules contribute to the observed sum intensity: linear and branched, saturated and unsaturated chains (code: s), cyclic molecules (code: c), and aromatic species (code: a). Note that aromatic molecules are also cyclic but aromaticity is prioritized and no double-assignments are made. Based on the relative abundance estimates corresponding to the fragment sum of the individual pure hydrocarbon molecules in

**Table 1 List of molecules identified in the coma of 67P on 3 August 2015.**

| # | Type | Molecule | Sum formula | HDI | Fragment sum | Error[a] | Previously detected |
|---|---|---|---|---|---|---|---|
| 1 | s | Methane | $CH_4$ | 0 | 10.7 | 3.5 | yes |
| 2 | s | Acetylene | $C_2H_2$ | 2 | 6.3 | 2.1 | yes |
| 3 | s | Ethylene | $C_2H_4$ | 1 | 1.7 | 0.56 | no |
| 4 | s | Ethane | $C_2H_6$ | 0 | 14.0 | 4.6 | yes |
| 5 | c | Cyclopropene | $C_3H_4$ | 2 | 2.6 | 0.86 | no[b] |
| 6 | s | Propene | $C_3H_6$ | 1 | 3.0 | 0.99 | no |
| 7 | s | Propane | $C_3H_8$ | 0 | 2.8 | 0.92 | yes |
| 8 | c | Cyclobutene | $C_4H_6$ | 2 | 1.1 | 0.36 | no[c] |
| 9 | c | Cyclobutane | $C_4H_8$ | 1 | 1.9 | 0.63 | no[c] |
| 10 | s | Butane | $C_4H_{10}$ | 0 | 2.1 | 0.69 | yes |
| 11 | c | Cyclopentadiene | $C_5H_6$ | 3 | 0.1 | 0.03 | no[b] |
| 12 | c | Cyclopentane | $C_5H_{10}$ | 1 | 2.8 | 0.92 | no[b] |
| 13 | s | Pentane | $C_5H_{12}$ | 0 | 1.5 | 0.50 | yes |
| 14 | a | Benzene | $C_6H_6$ | 4 | 2.4 | 0.79 | yes |
| 15 | c | Cyclohexane | $C_6H_{12}$ | 1 | 1.2 | 0.40 | no[b] |
| 16 | s | Hexane | $C_6H_{14}$ | 0 | 1.1 | 0.36 | yes |
| 17 | s | Isohexane | $C_6H_{14}$ | 0 | 1.8 | 0.59 | no[b] |
| 18 | a | Bicyclo[4.1.0]hepta-1,3,5-triene | $C_7H_6$ | 5 | 0.2 | 0.07 | no[b] |
| 19 | a | Toluene | $C_7H_8$ | 4 | 1.1 | 0.36 | yes |
| 20 | c | 1,3-Cycloheptadiene | $C_7H_{10}$ | 3 | 1.0 | 0.33 | no[b] |
| 21 | c | Methylcyclohexane | $C_7H_{14}$ | 1 | 2.4 | 0.79 | no[b] |
| 22 | s | Heptane | $C_7H_{16}$ | 0 | 1.0 | 0.33 | yes |
| 23 | a | Styrene | $C_8H_8$ | 5 | 0.1 | 0.03 | no[c] |
| 24 | a | p-Xylene | $C_8H_{10}$ | 4 | 1.4 | 0.46 | tentative[b] |
| 25 | c | 3-Ethenylcyclohexene | $C_8H_{12}$ | 3 | 1.3 | 0.43 | no[b] |
| 26 | c | 1,2-Dimethylcyclohexene | $C_8H_{14}$ | 2 | 1.4 | 0.46 | no[b] |
| 27 | c | 1,1-Dimethylcyclohexane | $C_8H_{16}$ | 1 | 0.1 | 0.03 | no[c] |
| 28 | c | Ethylcyclohexane | $C_8H_{16}$ | 1 | 1.4 | 0.46 | no[b] |
| 29 | c | Cyclooctane | $C_8H_{16}$ | 1 | 1.4 | 0.46 | no[b] |
| 30 | s | 2,5-Dimethylhexane | $C_8H_{18}$ | 0 | 0.2 | 0.07 | no[b] |
| 31 | s | Octane | $C_8H_{18}$ | 0 | 0.6 | 0.20 | tentative[b] |
| 32 | a | Indene | $C_9H_8$ | 6 | 0.1 | 0.03 | no[b] |
| 33 | a | Indane | $C_9H_{10}$ | 5 | 0.2 | 0.07 | no[b] |
| 34 | a | Mesitylene | $C_9H_{12}$ | 4 | 0.7 | 0.23 | no[b] |
| 35 | c | Octahydro-1H-indene | $C_9H_{16}$ | 2 | 0.9 | 0.30 | no[b] |
| 36 | c | 1,2,3-Trimethylcyclohexane | $C_9H_{18}$ | 1 | 0.7 | 0.23 | no[b] |
| 37 | s | Nonane | $C_9H_{20}$ | 0 | 1.0 | 0.33 | no |
| 38 | s | 2-Methyloctane | $C_9H_{20}$ | 0 | 0.8 | 0.26 | no[b] |
| 39 | a | Naphthalene | $C_{10}H_8$ | 7 | 0.7 | 0.23 | tentative |
| 40 | a | 1,2-Dihydronaphthalene | $C_{10}H_{10}$ | 6 | 0.4 | 0.13 | no[c] |
| 41 | a | 2,3-Dihydro-2-methyl-1H-indene | $C_{10}H_{12}$ | 5 | 0.2 | 0.07 | no[c] |
| 42 | a | 1,2,3,4-Tetrahydronaphthalene | $C_{10}H_{12}$ | 5 | 0.1 | 0.03 | no[c] |
| 43 | a | 1,4-Diethylbenzene | $C_{10}H_{14}$ | 4 | 0.1 | 0.03 | no[b] |
| 44 | c | Decahydronaphthalene | $C_{10}H_{18}$ | 2 | 1.6 | 0.53 | no[b] |
| – | a | Furane | $C_4H_4O$ | 3 | – | – | no[b] |
| – | c | Dihydrofuran | $C_4H_6O$ | 2 | – | – | no[b] |
| – | c | Tetrahydrofurane | $C_4H_8O$ | 1 | – | – | no[b] |
| – | a | Benzaldehyde | $C_7H_6O$ | 5 | – | – | no[b] |
| – | a | Benzylalcohol | $C_7H_8O$ | 4 | – | – | no[b] |
| – | s | Propanoic acid | $C_3H_6O_2$ | 1 | – | – | no[b] |
| – | a | Benzoic acid | $C_7H_6O_2$ | 5 | – | – | tentative |
| – | c | Pyrrolidine | $C_4H_9N$ | 1 | – | – | no[b] |
| – | a | Dimethylpyrrole | $C_6H_9N$ | 3 | – | – | no[b] |
| – | a | Benzonitrile | $C_7H_5N$ | 6 | – | – | no[b] |
| – | a | Benzylamine | $C_7H_9N$ | 4 | – | – | no |
| – | c | Dihydrothiophene | $C_4H_6S$ | 2 | – | – | no[c] |
| – | s | Methylhydrogendisulfide | $CH_4S_2$ | 0 | – | – | no[b] |

[a]Errors of the fragment sum (33%) for a specific molecule are smaller than the errors of the real abundance as these contain additional uncertainties of the ionization cross-section data, fragmentation data, and detector yields.
[b]Other isomers with available reference data are less likely.
[c]Other isomers are possible.
Pure hydrocarbon molecules (top part of the table) were identified from the Occam's razor[27] conform deconvolution of the respective sum intensity. Heteroatom-bearing molecules (bottom part of the table) were identified on the basis of strong and characteristic signals. Molecules are typecast according to their structural elements as chain-based (s), cyclic (c), and aromatic (a) species. Previous detections in comet 67P, as reviewed in Table 4 in Ref. [1], as well as the fragment sums (only available from deconvolution) and hydrogen deficiency index (HDI) values are indicated.

Table 1 an approximate ratio of 6:3:1 is obtained. This ratio estimate has a 1σ uncertainty of 10%, which includes statistical as well as systematic uncertainties. The relative abundance estimates may not well capture the effective abundance of specific molecules due to missing data on the ionization cross section, DFMS fragmentation pattern, and molecule-dependent detector sensitivity. However, their sums more accurately describe the relative abundance of groups of species. This is because isomers often expose similar fragmentation patterns and each of the three groups has characteristic signals that cannot be explained by representatives of the other groups.

Generally, we state that ring-based structures make up an important share of cometary complex organic molecules. Some of the aromatic and non-aromatic rings are substituted, mostly by short alkyl chains or functional groups like carboxylic acids (benzoic acid), amines (benzylamine), and possibly nitriles and alcohols (benzonitrile, benzylalcohol). Also, five-membered heterocycles are very likely present in the form of furane-, pyrrole-, and thiophene-derivatives. Whether heterocyclic six-membered rings also contribute is more difficult to establish and topic of extended investigations.

**Absence of polymers**. Based on these results, the controversy of polymeric matter versus individual molecules is now resolved. Although formaldehyde ($CH_2O$), hydrogen cyanide (HCN), and ammonia ($NH_3$), three abundant cometary molecules and high-ranking candidates for the formation of polymeric structures, are present on 3 August 2015 like at most other times, there are no signatures of their multimers. In particular, polymerized formaldehyde, or polyoxymethylene ($(CH_2O)_n$), was suggested to explain the PICCA spectrum obtained at comet 1P/Halley[13,14] and, the Ptolemy spectrum of comet 67P, collected 20 min after initial touch-down of Rosetta's lander Philae. Wright and co-workers[29] argued that the Ptolemy data would resemble the PICCA data and assigned several signals to POM decomposition products, among others the strong signature of the linear formaldehyde trimer on $m/z = 91$. High-resolution DFMS data in Fig. 2a, however, shows that the H-terminated linear formaldehyde trimer ($C_3H_7O_3$) on $m/z = 91$ is absent. The strong signal stems from the tropylium ion ($C_7H_7$), mostly due to toluene ($C_7H_8$; mw = 92 Da). Also, the intensity distribution of the two Ptolemy signals, $m/z = 92$ and 91, is in line with the fragmentation of toluene. The formaldehyde dimer is an isomer of acetic acid, a common cometary molecule[1,9], and therefore cannot be used to trace POMs in mass-spectrometric data. Calibration measurements with commercially available POMs on the DFMS laboratory twin model have been presented previously[30]. Paraformaledhyde ($OH(CH_2O)_nH$; $n = 8$–100) was thermally desorbed in the experimental chamber at temperatures lower than 40 °C. The fragmentation pattern, compatible with the reference pattern available from the National Institute of Standards and Technology (NIST)[31], is shown in Fig. 7 of Ref. [30]. The same publication demonstrates that high-resolution DFMS data from a dust grain entering the instrument's ion source show no evidence for the existence of POMs at comet 67P. Finally, hexamethylenetetramine (HMT, $C_6H_{12}N_4$; mw = 140 Da), a condensate of six formaldehyde and four ammonia molecules, which would show a strong M signal on $m/z = 140$, is not detected on 3 August 2015, cf. Fig. 2b. This molecule was intensively studied as a promising candidate for the observed distributed source of the cyano radical (CN), see Ref. [32] and other work referenced therein. While we show comet 67P's neutral coma—close to perihelion at a cometocentric distance of roughly 215 km—to be rich in complex organics similar to those suggested for comet 1P/

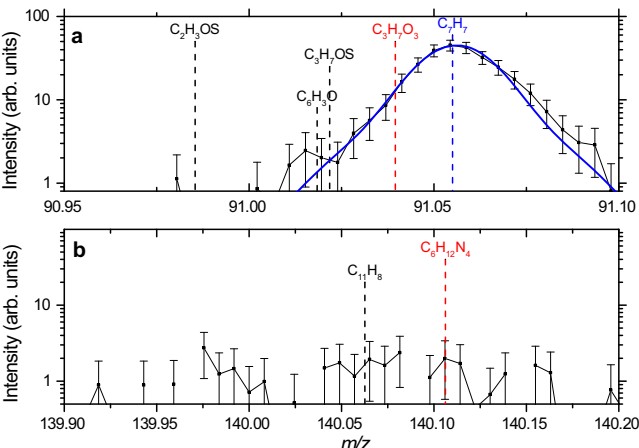

**Fig. 2 Mass spectra from 3 August 2015 at the exact masses of two typical multimers.** Intensities measured by the Double Focusing Mass Spectrometer (DFMS) are given in arbitrary units (arb. units) with error bars representing 1σ statistical errors. **a** Fitting a double Gaussian peak function (blue line) as detailed in the Methods section shows that the H-terminated linear trimer fragment of polyoxymethylene ($C_3H_7O_3$) is absent or within error margin on $m/z = 91$. The prominent signal corresponds to the tropylium ion ($C_7H_7$), mostly originating from the fragmentation of toluene ($C_7H_8$). **b** On $m/z = 140$, the pure hydrocarbon position is separable from the molecular ion signal of hexamethylenetetramine ($C_6H_{12}N_4$). Both signals are absent within error margins. The positions of the exact masses are indicated with dashed lines.

Halley[16,17], polymeric matter in the form of POMs and HMT is, if present, of extremely low abundance.

**$sp^2:sp^3$ and $CH_2:CH_3$ ratio**. A way to express the local geometry of a carbon atom in a molecule is via its hybridization state. Carbon atoms with s and p orbitals are said to be sp-hybridized when they are in a linear (triple-bond) environment, $sp^2$-hybridized when they are in a trigonal-planar (double-bond) environment, and $sp^3$-hybridized when they are in a tetrahedral (single-bond) environment. The average $sp^2:sp^3$ ratio, therefore, characterizes a set of molecules structurally. This ratio is derived experimentally, e.g., from infra-red spectroscopy, or calculated by weighted counting of the number of correspondingly hybridized C atoms. For the set of pure hydrocarbon molecules observed by the DFMS, cf. Table 1, the average $sp^2:sp^3$ ratio is ~0.3 with respect to the C atoms and ~0.1 with respect to the H atoms. Because there are relatively little sp-hybridized C atoms (only the two C atoms of acetylene are securely identified), roughly one third of the C atoms are involved in double-bonds (aliphatic or aromatic). Comparing these values to available literature demonstrates surprising similarities with interstellar carbonaceous dust. Dust features towards the extragalactic infra-red galaxy IRAS 08572 + 3915 allowed to observationally constrain the $sp^2:sp^3$ ratio related to the H atoms to ≤0.08[33]. Together with Random Covalent Network model calculations, these constraints helped the authors of Ref. [33] identify the observed hydrocarbons as polymer-like hydrogenated amorphous carbonaceous matter dominated by olefinic backbone structures. It was suggested that subsequent energetic processing could lead to elimination of hydrogen atoms and, thus, to a higher $sp^2:sp^3$ ratio and likely an increased aromaticity. Assuming the average ratio derived in this work for the dominant group of pure hydrocarbon molecules to be representative for 67P's organics, cometary complex organics and interstellar carbonaceous matter expose consistent structural characteristics.

Another characteristic parameter of (macro)molecular matter is the content of $CH_2$ with respect to $CH_3$ – the $CH_2:CH_3$ ratio. Keller et al.[34] reported cometary dust returned to Earth in the framework of the Stardust mission to expose a ratio of ~2.5. These authors compared their findings to other reservoirs of organics, citing the most relevant literature: While interplanetary dust particles (IDPs) expose ratios of ~2.5 comparable to comets, both carbonaceous matter in meteorites and the diffuse ISM clearly fall below the cometary values with ~1.5 and 1.1–1.3, respectively. 67P's pure hydrocarbon species of Table 1 yield an approximate $CH_2:CH_3$ ratio of ~1.9. Thus, cometary dust, as well as IDPs, contain more extended and/or less branched aliphatic moieties than meteoritic matter or the diffuse ISM. Differences are thought to arise from different environmental conditions entailing different material processing. Laboratory experiments successfully produced hydrogenated amorphous carbon polymers from photolysis of a series of organic molecules at low temperatures[35]. The produced films exposed an olefinic $CH_2$ to aliphatic $CH_3$ ratio of 0.05–0.1, which is on the same order of magnitude as for the cometary ensemble reported here, while accompanying modelling work showed absorption bands of the diffuse ISM to be better reproduced based on macromolecular structures containing less extended aromatic systems as compared to the aliphatic body. In 67P's organic inventory, extended aromaticity cannot be observed either.

**Average composition**. The average composition of the species associated to the mass-spectrometric data set presented in Fig. 1 is a direct way to characterize the full volatile organic budget of comet 67P as it does not necessitate deconvolution into individual fragmentation pattern or molecules. It can be approximated by multiplying the stoichiometric coefficients of the individual species' formula with the signal intensities, summing the resulting values per element, and normalizing the sums to carbon. To describe 67P's budget of volatile complex organic molecules most accurately, it is appropriate to exclude from the analysis the abundant and C-bearing (but inorganic) volatiles, namely, $CO$, $CO_2$, $HCN$, $CS_2$ and $COS$, cf. Methods section for further information. The average sum formula of four- or more-atomic C-bearing neutral species corresponds to $C_1H_{1.56}O_{0.134}N_{0.046}S_{0.017}$. The stoichiometric coefficients are listed in Table 2 together with selected literature.

The results of this work and the findings of the COSIMA secondary-ion mass spectrometer[36], which analysed ejected cometary particles after collection, storage, and microscopic pre-analysis, are compatible: Organic volatiles contained in the dust have likely sublimated during the processes mentioned above and prior to the compositional analysis. For the roughly 45% organic fraction of the particles[4], which were reported to be of macromolecular structure[10], only the H/C and the N/C ratios have been measured. As oxygen is present in both the mineral and the organic phase, the O/C ratio was estimated from a set of assumptions[4] and thus has a high uncertainty. Sulphur was assumed by the same authors to be mostly present in the form of iron sulphides. While COSIMA's N/C ratio derived from a sample of 27 particles[37] is nearly identical to our value, the H/C ratio derived from a sample of 33 cometary particles[38] is clearly lower. This is interpreted in terms of a pronounced hydrogen deficiency in the refractory organics as compared to the volatile ensemble observed by the DFMS.

Interestingly, the heteroatomic abundances relative to carbon seem to be conserved in organic materials beyond comet 67P. The PUMA mass spectrometer on the Vega1 spacecraft analysed comet 1P/Halley's dust particles impacting the instrument with a relative velocity of over 77 km/s[17]. The organic phase of the

**Table 2 Average composition of organic species observed on 3 August 2015.**

| Sample | H/C | O/C | N/C | S/C | References |
|---|---|---|---|---|---|
| 67P, Rosetta/DFMS (organic coma species) | 1.56 ± 0.04 | 0.134 ± 0.007 | 0.046 ± 0.003 | 0.017 ± 0.001 | This work |
| 67P, Rosetta/COSIMA (average of different samples of particles) | 1.04 ± 0.16 | ~0.3 | 0.035 ± 0.011 | | 38, 4, 37 |
| 1P/Halley, Vega1/PUMA (organic mantle) | 0.8 (error unknown) | 0.2 (error factor 2) | 0.04 (error factor 2) | 0.02 (error factor 2) | 17 |
| 81P/Wild 2, Stardust (6 dust tracks) | | 0.16–0.56 | 0.02–0.21 | | 20 read from Fig. 3(B). |
| IOM (estimate for CI, CM, and CR class meteorites) | 0.7–0.8 | 0.15–0.20 | 0.03–0.04 | 0.01–0.04 | 39 |
| SOM (Murchinson meteorite) | 1.55 | 0.20 | 0.03 | 0.03 | 40 |
| Saturn's ring rain, Cassini/ INMS (organics without $CH_4$) | ~1.6 (1.9–2.3) | ~0.1 (0.05–0.18) | ~0.15 (0.01–0.1) | negligible | 41 read from Fig. 2 (max. frequency value for auto-fits, range for hand-fits) |

The average elemental abundance ratios relative to carbon are listed together with selected reference data of comets, meteorites-insoluble and soluble organic matter (IOM and SOM) –, and Saturn.

particles, thought to form a mantle around a mineral core, seems to expose a very similar composition like 67P's complex organics ensemble analysed in this work, with relatively large uncertainties though. In contrast, Stardust samples, collected during a flyby at comet 81P/Wild 2 and returned to Earth for analysis, show O/C and N/C ratios that tend to be slightly higher and exhibit a large scatter[20].

The organics contained in meteorites are divided into a soluble and an insoluble part, soluble organic matter (SOM) and insoluble organic matter (IOM). These two classes expose very similar heteroatomic abundances but quite different H/C ratios[39]. Also, for meteoritic matter, heteroatom relative abundances are similar to those in comet 67P's organics as seen by COSIMA and the DFMS as well as comet 1P/Halley's organic dust. Interestingly, the H/C ratio for SOM extracted from the Murchison meteorite of 1.55[40] is in perfect agreement with the value we report here. Notably, the CI, CM (e.g., Murchinson), and CR classes of meteorites are on the least-processed end of the scale while the H/C ratio is expected to decrease with increasing degree of processing mostly due to impact heating. From our analysis, complex cometary organics are very similar to meteoritic SOM—almost identical—even regarding the degree of hydrogen deficiency and likely also the size of the molecules.

Moreover, the elemental abundance ratios reported by Miller et al.[41] for the organics of Saturn's ring rain material are a good match with the values we report in this article. The Cassini instrument collecting the underlying data, INMS, is an EII mass spectrometer like the DFMS, but with unit-mass-resolution only. Elemental abundance ratios were extracted from data deconvolved on the basis of fits—performed either by hand or automatically—and are subject to various sources of uncertainties as discussed in detail by these authors. The best agreement with our values is found for the organics when methane ($CH_4$), which interplays with fragments of the larger organics, is excluded from the fitting process. This exclusion primarily affects the auto-fits while the hand-fits produce similar results in both cases. Our N/C value is situated on the low end of the hand-fit range, which is due to the fact that we have excluded HCN, a major N-bearing volatile, from our analysis. Inclusion of HCN would lift our N/C ratio by roughly a factor 2 and lead to a good agreement with especially the hand-fitted values from Ref. [41]. In the light of the yet unclear origin of Saturn's ring material, this surprising match between the organic part of Saturn's ring rain material and cometary complex organics boosts the hypothesis that the two material reservoirs may be of a similar origin. As of today, two concurring scenarios of origin exist:[42] recent formation due to (1) a collisional disruption of an icy moon (e.g.[43,44],) or (2) tidal splitting of a passing comet (e.g., Ref. [43]) most likely during the Late Heavy Bombardment versus early formation during the times when the Solar System formed. However, also evolutionary processes occurring in the ring system, such as incoming meteoroid material composed of a silicate and an organic phase, must be taken into account[42]. A future detailed comparative analysis of Cassini and Rosetta data, which is beyond the scope of the current work, will provide further insights.

**Evidence for a pre-solar origin**. Having observed a pronounced consistency of the heteroatom abundances relative to carbon in the different reservoirs of organics surveyed in Table 2, as opposed to a high variability of the degrees of hydrogenation, we here argue that this is consistent with a shared pre-solar history of Solar System organics and subsequent material processing. We suggest to picture cometary organics, available to present day in-situ studies, to be the outcome of a long sequence of material processing, starting in the ISM.

In the ISM, two relevant and opposing scenarios of molecule-formation, namely top-down versus bottom-up, are discussed. Ref. [45] and literature cited therein presented an extensive dust evolution framework showing how nascent amorphous carbon nanoparticles, interacting with their ISM environment, could develop heteroatom-bearing chemical structures at their surfaces which are locally similar to those observed in the ISM but also later in comets. However, Dartois and team report similar structures from top-down polymerization chemistry in the laboratory[35]. For the case of PAHs, which are the most complex individual organic molecules detected in the ISM to date, it is heavily debated whether formation from sequential reactions of smaller molecules (bottom-up) or from degradation/processing of larger molecular entities (top-down) is more likely. In the laboratory, the degree of hydrogenation of carbonaceous starting material has been pushed both ways and bottom-up as well as top-down formation routes of PAHs appear feasible[46,47]. Although the top-down argumentation was favoured by the community for years in order to explain the existence of large PAHs and fullerenes, the recent detection of benzonitrile in the Taurus Molecular Cloud 1 (TMC-1)[48] pushed the bottom-up scenario. According to modelling work of Lee and team[49], bottom-up formation is a viable alternative especially for smaller aromatic species. Undoubtedly, it is a long way from primordial matter in the ISM towards evolved solar systems, like ours, and there are processes along this way which could decrease the H/C ratio—and accordingly also the hybridization states of the C atoms—of the original organic matter (e.g., UV-processing) and others that could do the opposite (e.g., H atom bombardment). Our analysis of cometary organics presented here led to the identification of several aromatic molecules and their hydrogenated derivatives as can be seen from Table 1 (e.g., benzene <-> cyclohexene <-> cyclohexane or naphthalene <-> dihydro-naphthalene <-> tetrahydronaphthalene <-> decahydronaphthalene). These molecules are likely present in 67P's coma simultaneously, which could be the result of a hydrogenation or dehydrogenation process having occurred at some point in time prior to detection, most likely prior to comet formation.

Although energetic processing of material is possible during and after comet formation, e.g., due to decomposition of refractory organic matter on expelled and heated-up dust grains, evidence is accumulating that pre-solar inheritance of cometary complex organics is likely. In the following, we summarize the arguments that apply to the case of 67P's organics:

1. Comparative work done for the comets 1P/Halley[50] and 67P[51] suggests that cometary ices tightly resemble ices in the ISM regarding their composition, despite the fact that comets are evolved objects.

2. First-of-its-kind modelling work indicates that chemical reactions during this cold storage phase are very limited[52]. Exposed to radiation, the surface layer is found to be most affected. However, even a weakly active comet like 67P loses several tens of centimetres of material during a single perihelion passage[53]. As 67P has approached the Sun many times on its current orbit, effects of material processing during cold storage are expected to be irrelevant.

3. Some molecules of the organic ensemble reported here, for instance benzene, were observed by the DFMS during the full Rosetta science mission, at heliocentric distances as far as >3.8 au and at cometocentric distances as close as 2.5 km[8]. These molecules are therefore mostly related to ice-sublimation rather than to dust activity. Their detection should be correlated in a prominent way with the comet's dust activity would they efficiently form from decomposing refractory organics.

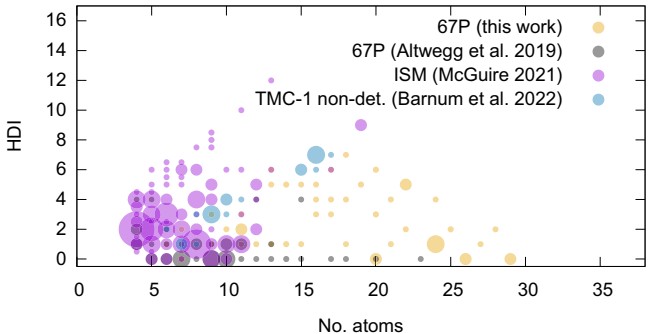

**Fig. 3 Comparison of the cometary ensemble to the Interstellar Medium (ISM).** C-bearing neutrals, composed of four atoms or more, identified in the coma of comet 67P previously (Ref. [1] grey) or newly in this work (yellow) are compared to those detected in the ISM (Ref. [60]; purple). Furthermore, non-detection (non-det.) of a series of heterocycles in the Taurus Molecular Cloud 1 (TMC-1) are included (Ref. [65]; blue). The molecules' hydrogen deficiency index (HDI) values are plotted as a function of the number of constituting atoms (no. atoms). The size of the circles represents the number of molecules sharing HDI value and number of atoms (the smallest circles correspond to one molecule).

4. Analysis of DFMS data recently showed that the D/H ratio of the first four linear alkanes in 67P's inner coma is more than four times higher than that of cometary water[54] and compatible with the D/H ratio derived for 67P's refractory organics from COSIMA data[55]. The D/H value for 67P's methanol was even higher[56]. If complex organics were formed in-situ, then they should exhibit D/H values compatible with the Solar value or at least with cometary water. Hence, high D/H values support the inheritance scenario.

5. Relevant fragmentation of macromolecular refractory organic matter must go alongside with hydrogenation in order to reproduce the observed H/C ratio. Both possible sources of hydrogen, solar wind and photodissociation of cometary water, seem unfavourable as they would lead to D/H values inconsistent with observations. Moreover, solar wind is well shielded by the comet's activity around perihelion[57] and the dissociation of water is too slow to be relevant at cometocentric distances of 215 km at perihelion (the lifetime of the reaction $H_2O \rightarrow OH + H^+ + e^-$ equals to $\sim 10^7$ s at 1 au[58]).

6. In-situ neutral-ion chemical reactions are playing a subordinate role in 67P's tenuous gas phase. This was convincingly demonstrated for the case of water at perihelion[59]. Complex organic molecules cannot be produced efficiently that way.

These arguments well support the conclusion that the complex organic molecules of the reported cometary ensemble are likely not the product of material processing on the comet but are rather inherited from earlier stages of the Solar System history and witness to an organic chemistry from dark clouds to comets that is far more complex than previously thought. Comparing and contrasting the cometary and ISM censuses of organics can provide useful insights into the differences and similarities of the two reservoirs. Figure 3 visualises, based on their HDI values (HDI = $C + 1 + N/2 - H/2 - X/2$, where $C$ is the number of C atoms in the molecule, $N$ the number of N atoms, $H$ the number of H atoms and $X$ the number of halide atoms), C-bearing molecules composed of at least four atoms identified in comet 67P to date (Table 4 of Ref. [1] and Table 1 of this work) and those detected in the ISM as of 1 June 2021[60]. The HDI compares

the number of H atoms in a given molecule to the maximum number of H atoms and, hence, bears information on the molecule's chemical structure as it corresponds to the total number of multiple C-C bonds and cycles in the molecule. From Fig. 3, the overlap between the ISM and the cometary census appears rather limited in terms of HDI versus number of atoms. ISM molecules tend to be smaller and more hydrogen deficient than cometary species. While some of these trends are expected to be real, others may be caused by methodological biases and limitations:

1. Ground-based observations of molecular clouds, prestellar cores, and star-forming regions are predominantly carried out using rotational spectroscopy, which is sensitive to molecules in cold environments. Rotational spectroscopy is essentially limited to molecules with a permanent electric dipole moment. Symmetric molecules can therefore not be probed. On the other hand, rotational spectroscopy is able to distinguish structural isomers.

2. For the technique of mass spectrometry, in contrast, a molecule's dipole moment is not relevant but only its exact mass. However, even at the high-mass resolution of the DFMS, mass spectrometry is not capable of distinguishing between structural isomers unless these molecules show significantly different fragmentation patterns, as detailed in the Methods section.

3. Environmental parameters like temperature do not only determine which chemical species preferably form but also which can survive in a gaseous state. Should molecules form, e.g., on a grain surface, they need to leave this surface in order to be probed by either experimental technique under discussion, rotational-spectroscopy or mass spectrometry.

4. Moreover, both methods rely on laboratory reference spectra. Molecules for which no reference data is available can usually not be securely identified. For the case of mass spectrometry, further information is presented the Methods section.

It is clear that the ISM hosts diverse astrophysical environments, including environments that are very different from comets, and molecules listed in the ISM census[60] may not be present in all of them. Nevertheless, and irrespective of the methodological biases and limitations mentioned above, both perspectives—from observations with radio- and other spectroscopic techniques and in-situ mass spectrometry—complement one another in order to establish a comprehensive picture of the composition and compositional evolution of cosmomaterials. This can be illustrated on the basis of a few specific examples related to Fig. 3. While the ISM accommodates linear (but functionalized) polyynes composed of relatively few atoms but with relatively high HDI values, the comet apparently does not. Our mass-spectrometry-based work failed to identify polyynes of the general structure $C_nH_2$, where $n > 2$ indicates the number of carbon atoms in the molecule, due to the lack of reference data. However, observed upper limit intensities may be assigned to the corresponding M, cf. Supplementary Fig. 1. Yet another striking difference is the absence of radicals in the comet, which, however, may have similar reasons. We did not systematically include radicals in our analyses up to now, because there are usually no mass-spectrometric reference data available for such highly reactive species. This does not mean that radicals cannot be detected with our instrument. The presence of the CN radical in 67P's inner coma has recently been reported by Hänni and co-workers[61]. Based on the unexplained intensity contained in Supplementary Fig. 1, it may be possible to establish a tentative contribution of coma radicals, e.g., of the methyl radical ($CH_3$) or

larger species. This exceeds the scope of this work and is a topic of future investigations. Vice versa, radio- and (sub-)millimetre spectroscopic studies of the ISM miss symmetric species like, for instance, the linear alkane chains we report here. Also, the long sought-after, most simple representative of the class of polycyclic aromatic hydrocarbons (PAHs), naphthalene, has remained undetected in the ISM so far. While CN-substituted derivatives (e.g., cyano-naphthalene[62]) can be used as proxies for species which do not possess a dipole moment in their unfunctionalized form, our work allows to directly observe symmetric species. We confirm the previously tentative identification of unfunctionalized naphthalene in comet 67P[1] and report several naphthalene-derivatives, including the fully hydrogenated specimen, to be likely simultaneously present in 67P's coma, although of generally low abundance. We also find cyclopentadiene, of which two CN-derivatives have been identified in TMC-1[63,64]. However, there are also apparent differences. A recent search for a series of heterocyclic candidate molecules in TMC-1, using a deep, broadband centimetre-wavelength spectral line survey of the region, has yielded upper limits only[65]. The targeted molecules are included in Fig. 3. It is interesting to note that they are located in the boundary region between the identified ISM and cometary species. Arguments for and against the existence of these heterocycles in dark clouds are brought forth in Ref. [65] and the cited literature. Now, the ongoing debate can be supported with additional evidence from our comet study pointing at the presence of heterocycles (e.g., furane or a pyrrole-derivative in Table 1).

Looking out to possible future efforts in the fields of astrochemistry and solar system science, we do not only propose to consult comet studies for guiding the identification of promising candidates for ISM molecular searches but, vice versa, to take such searches as a motivation for detailed analyses of in-situ cometary and possibly other Solar System data. Apart from revisiting our data in search of polyynes, radicals, and other small or hydrogen-deficient species, we plan to direct future work on the unique high-resolution mass-spectrometric data from ESA's Rosetta mission towards heteroatom-bearing species, including heterocycles.

## Methods

**Instrumentation and data reduction**. ROSINA's high-resolution double focusing mass spectrometer (DFMS) contains an ion source, a toroidal shaped electrostatic analyser, and a magnetic sector in Nier-Johnson configuration, which separate the ionized coma species (parents and charge-sustaining fragments) according to their ratio of mass ($m$) and charge ($z$). A complete description of the instrument is given in Balsiger and co-workers[22]. Here we highlight some instrument characteristics which are crucial for this work. Since we investigate cometary neutrals here, for simplicity, we omit indication of the charged state of the registered species.

The mass resolution of the DFMS is equal to 3'000 at 1% peak height of the peak at $m/z = 28$, decreasing for higher masses. Consequently, the detector covers less than 1 $m/z$ below approximately $m/z = 70$ and the instrument has to step through the nominal $m/z$ range from 12 to 180 by adjusting the energy of the ions. For $m/z > 70$, an additional post acceleration potential of 1000 V in front of the detector increased the detection efficiency.

Data in this work were collected on 3 August 2015, using a combination of measurement modes (222, 562, and 564). Specifically, $m/z = 44$ to 80 were measured between 15:03–15:20 UTC, $m/z = 80$–140 between 15:33–16:00 UTC, and $m/z = 13$–43 between 16:25–16:39 UTC. The data reduction procedure for DFMS data has been described elsewhere, e.g., in Ref. [5]. A representative example of a DFMS high-resolution mass spectrum evaluated for this paper is shown in Supplementary Fig. 2. The individual peak areas are extracted from fits of the sum spectra over both detector rows. Corrected for the instrument's sensitivity, a function of the species' mass. These areas correspond to the number of impinging ions in arbitrary units as used in this work. It is impossible to derive effective ion numbers without knowing the species responsible for the peak as the response of the MCP detector depends on the structure of the species.

Measurements have several sources of uncertainty. For $m/z < 100$ counting statistics is a minor contribution. For $m/z > 100$ the statistical uncertainties are ~20%. All measurements were done on the highest gain step, meaning detector gain uncertainty cancels out for relative abundances. The fit error from

disentangling different peaks in the same spectrum and the individual pixel gain error was estimated to be ~15%. The drift in density over the time scale of the data collection adds to the uncertainty. Such drift can have several reasons: (1) spacecraft slewing, (2) comet rotation, (3) variability of coma composition. For the data set considered here, all of this occurred simultaneously, amounting to a considerable distortion of the data. The total neutral pressure, as measured by ROSINA's COmet Pressure Sensor (COPS; cf. Ref. [22]), is commonly used for drift corrections. However, due to COPS' much larger field of view as compared to the DFMS, it cannot be used to correct for density changes due to slewing. A better correction is achieved by comparing overlapping mass ranges. For instance, $m/z = 44$–79 was measured twice (15:03–15:19 and 16:40–16:56 UTC) as was $m/z = 80$–100 (15:20–15:28 and 15:33–15:42 UTC). No drift was observed for the mass range $m/z = 80$–100. There was an almost linear drift of 20% ($m/z = 44$) to 50% ($m/z = 79$) for the mass range $m/z = 44$–79. This was then extrapolated backwards to correct the $m/z = 12$–43 intensities. However, there remains some systematic uncertainty of 15%. The resulting total uncertainty is 30% for $m/z < 100$ and 50% above. Due to the high variability of the observational conditions as indicated above, stacking of mass spectra does not result in a reduction of uncertainty but it actually increases it.

**EII fragmentation**. Neutral coma species investigated in this work were ionized in the DFMS's ion source through electron impact ionization (EII) prior to separation according to their $m/z$. In the EII process, fragmentation of the parent species occurs inside the mass spectrometer and leads to a characteristic fingerprint for each thus analysed molecule, the so-called EII fragmentation pattern. Usually, the molecular ion (M) corresponds to the peak at the highest $m/z$ value (if iso-topologues are neglected). This peak, however, is not always the most intense peak. While aromatic species often produce strong molecular ion peaks, other species like certain hydrocarbon chains or primary alcohols or nitriles usually do not. For our analysis, we thus not only focused on the molecular ion peak, which is important for the secure identification of a molecule, but took into account the full frag-mentation pattern of a given species instead. Therefore, we relied on reference spectra either calibrated on the DFMS flight spare (mostly Refs. [8,9]) or available via the National Institute of Standards and Technology (NIST) mass spectrometry data base, part of the Chemistry Webbook 69[31]. Building on NIST reference data as compared to building on reference data calibrated with the DFMS itself introduces additional uncertainties for two reasons: First, NIST data is standardized to EII energies of 70 or sometimes 75 eV, while DFMS uses 45 eV. Minor differences in the fragmentation patterns occur, in particular, parent species tend to be slightly enhanced with respect to their fragments in the DFMS[8,9]. Second, NIST unit resolution standardized data of heteroatom-bearing species do not allow distinction between pure hydrocarbon fragments and heteroatom-retaining fragments should they share the integer mass, while DFMS does. Therefore, calibrated reference data was used if possible. Structural isomerism introduces additional uncertainties. This problem, however, is inherent to the technique of mass spectrometry. Complex organic molecules, as those targeted in this work, possess structural isomers by a number which grows rapidly with the number of atoms in the molecule. Isomers often show similar EII fragmentation patterns but isomer reference is data not necessarily available, hampering differentiation. In the case of the cometary coma, it is also possible that several isomers co-exist.

To minimize the uncertainties mentioned above, our detailed analysis is restricted to the subset of pure hydrocarbon species. This has the advantages that (1) all fragments must be pure hydrocarbons again and (2) all other groups of species (i.e., heteroatom-bearing species) are comparably less abundant and therefore do not contribute substantially to the pure hydrocarbon sum spectrum via fragments. Rare cases, where the heteroatom-bearing molecules are abundant and possess prominent pure hydrocarbon fragments, will be discussed in detail below. Minor isotopes were neglected.

**Derivation of average sum formula**. The average composition of the registered species (a mixture of molecular ions and charge-sustaining fragments of these molecular ions) is a proxy for the average composition of the ensemble of sample molecules. The considered data set was prepared to only include organic species. This means that we restricted our analysis to C-bearing species composed of four atoms or more. We also excluded the spacecraft fuel, methylhydrazine ($CH_6N_2$), and its fragments. Methylhydrazine was observed on rare occasions, e.g., when the spacecraft slewed and methylhydrazine, condensed in close vicinity of the thrusters, suddenly evaporated[66]. Also, the very rare and low-intensity group of NS-bearing organics were not taken into account as they do not influence the outcome sig-nificantly. In return, we included a contribution of fragment carbon ($^{12}C$), esti-mated to be $^{12}C = 99 \times {}^{13}C$ under the assumption of a terrestrial isotopic abundances[67]. The $^{12}C$ signal was lying outside the mass range scanned on 3 August 2015. Similar contributions of N, O, and S were neglected because heteroatom-bearing species are (1) less abundant than pure hydrocarbon species and (2) fragment rarely into the pure heteroatoms. Upper limits were derived for species with nearly identical exact masses, usually S-bearing and O-bearing hydrocarbon species. For the calculation of the average sum formula, always half of the upper limit signals were used.

The average sum formula was derived as follows: First, signals were multiplied with the stoichiometric coefficient of the assigned species, for each element

separately. Then, the weighted intensity was summed for each element. Finally, the sum intensity was normalized with respect to the sum intensity of carbon. The statistical error of the elemental abundances relative to carbon was calculated as $1\sigma$ error. For signals at $m/z \leq 100$ a 15% error was used for the error propagation calculation, while for signals at $m/z > 100$ a 35% error was used. Additional systematic errors may arise because (1) some fragments are more likely to retain the charge than others, (2) some molecular ions and fragments lie outside the analysed $m/z$ range, i.e., below 13 or above 140, and (3) the effective densities also depend on the EII cross-section of the probed molecules, which are species-dependent. It is expected that these effects cancel each other out to a large extent. Except, the H/C ratio is turned into a lower limit as we include an estimate of $^{12}$C but none of H, while both lie outside the measured $m/z$ range.

**Deconvolution of pure hydrocarbon sum spectrum**. To deconvolve the subset of pure hydrocarbon species, we follow an approach known as Occam's razor (e.g., Ref. [27]). Applied to our deconvolution problem, this heuristic principle demands the combination of as few different molecules as possible to explain as much of the observed intensity as possible. Although the solution obtained from such approach is not unique, especially as other isomers with similar fragmentation patterns can contribute, it is capable of capturing the ensemble's overall structural properties. The results (Supplementary Fig. 1 and Table 1 in the main text) demonstrate the simultaneous presence of three groups of molecules: hydrocarbon chains, cyclic molecules, and aromatic molecules. In the following, we discuss the details of the deconvolution separately for these three groups of species while referring to Supplementary Table 1. This table summarizes the reasons for inclusion of a given molecule and comments on the level of confidence with which this molecule was identified.

*Hydrocarbon chains*. It makes sense to start with filling in scaled reference intensities of the saturated hydrocarbon chains as the corresponding ions usually are molecular ions and not fragments. Saturated molecular ions are found to be present up to nonane. Decane, with M on $m/z = 142$, may be present as well but is not included as we restrict our analysis to species with molecular weights within the measured mass range. For octane and nonane, NIST reference spectra are employed as no calibration experiments have been performed with these compounds. Comparison of the calibrated fragmentation pattern of heptane to NIST shows that the DFMS, with only 45 eV electron energy, produces relatively more molecular ions, roughly 50% in that case. This could explain why inclusion of octane and nonane leads to overpopulation of the characteristic alkane chain fragments on $m/z = 43$ and 57. Inclusion of suitable branched (e.g., isohexane) and unsaturated species (e.g., ethylene and propene) improves the fit. However, quite a substantial amount of observed intensity still remains unexplained.

*Cyclic non-aromatic species*. The spectrum of non-aromatic cyclic species ranges from three-membered rings (cyclopropene) to bicyclic molecules (bicycloheptatriene). However, six-membered rings (cyclohexane) and their alkyl-chain-substituted derivatives (ethylcyclohexane and trimethylcyclohexane) as well as condensed cyclic systems (octahydro-1H-indene) seem to play an outstanding role. Although it may be possible that some of the alkyl-chain substituents were assigned to the wrong position in the molecule or that minor isomers were completely missed out, we can securely state that alkyl-chain-substituted non-aromatic cyclic species are present in our ensemble of molecules. The same argumentation applies to hydrogen-deficient rings: Although we may get the position of the carbon multiple bond in the system wrong (if multiple positions are possible) or miss out on minor or indistinguishable isomers completely, cyclic species with carbon multiple bonds contribute clearly to the observed intensity (cyclopropane, cyclopentadiene, etc.).

*Aromatic species*. The simplest aromatic moiety in the investigated ensemble of pure hydrocarbon molecules is the six-membered benzene ring. It is found in its pure form (benzene) as well as in various derivatives (toluene, styrene, xylene) and even in condensed form (naphthalene). Naphthalene is the simplest representative of the group of polyaromatic hydrocarbons (PAHs). Like other PAHs, naphthalene produces a strong molecular ion signal (on $m/z = 128$) and several groups of minor fragment signals. Unfortunately, the fragmentation signatures of higher-mass PAHs that fall below $m/z = 140$ are not sufficiently prominent and specific to allow for a clear identification. This means that the contribution of PAHs larger than naphthalene to the observed intensity distribution is unclear. It must be considered in addition that larger PAHs need more thermal energy to desorb and may thus sublimate less efficiently from the cometary dust. Moreover, aromaticity is observed in heteroatom-bearing species either in the form of heteroaromatic rings or benzene rings with heteroatom-bearing substituents. Details are discussed below.

*Residual intensity*. Summing up the selected pure hydrocarbon species, as done in Supplementary Fig. 1, still leaves some of the observed intensity unexplained, especially at $m/z > 100$. This can have several reasons: (1) The molecular ion producing an unexplained portion of intensity in the mass range may lay outside of the mass range, i.e., $m/z > 140$. Indeed, at various times of the Rosetta science phase, signals have been registered also for $m/z > 140$. As the corresponding

measurement mode (no. 566) has only been designed after August 2015 and has not been run frequently, these incidences are rare. Moreover, larger molecules need more thermal energy to sublimate and are thus expected to be rarely present in the coma far away from the Sun. (2) No reference mass spectra are available for certain groups of molecules. This is the case, for instance, for the group of linear polyynes of the form $H(C_2)_nH$, where $n > 1$. The nitrile-derivatives of such species, cyano-polyynes of the form $HC_nN$ ($n = 3, 5, 7$, etc.), which do possess a permanent dipole moment, have been observed in the ISM on a regular basis[60]. Whether or not linear polyynes are represented in the ensemble is, however, difficult to determine, given that the corresponding residual intensity is dominated by upper limit values. (3) As already stated above, both the uncertainties of the data as well as of the reference data have to be considered. Reference fragmentation patterns from NIST are clearly less reliable than those calibrated on the DFMS and likely to miss out on some of the molecular ion intensities. (4) Last but not least, heteroatom-bearing species with molecular ions within the scanned mass range may sometimes produce pure hydrocarbon fragments. Some of these fragments are identified thanks to strong and characteristic signals as detailed below.

**Identification of heteroatom-bearing species**. A deconvolution of the full data set, including all heteroatom-bearing species, is not possible. In addition to the already complex process and the uncertainties described above for the pure hydrocarbon molecules, the resolution of the NIST data becomes a problem: Unit resolution is not sufficient to distinguish between heteroatom-retaining and pure hydrocarbon fragments, should both fall on the same $m/z$ value. Relying on educated guesses (based on a fundamental understanding of EII-induced fragmentation) and on the fact that certain molecules exhibit very characteristic fingerprints, a few identifications are nevertheless possible. The newly identified heteroatom-bearing molecules, i.e., those that have not been identified in the coma of comet 67P previously[1], are listed in Table 1 in the main text as well as Supplementary Table 1 and are discussed in the following.

The subgroup of phenyl-derivatives R-CH$_2$-Ph, where R is a residue and Ph is the phenyl unit, is especially interesting to investigate. Thanks to their aromatic core, these molecules often exhibit a strong M signal which is sometimes even followed by a similarly strong M-H signal and a set of characteristic fragments. One of the possible fragments, $C_7H_7$ on $m/z = 91$, corresponds to the highly stable tropylium cation[68], which forms upon rearrangement of the ionization fragment CH$_2$-Ph. A typical example is toluene ($C_7H_8$), which was identified in the coma of comet 67P already by Schuhmann and co-workers[8] and is also represented in our data. In cases where the molecular structure does not allow formation of the tropylium cation, e.g., for R-CO-Ph, the phenyl cation $C_6H_5$ on $m/z = 77$ is observed. The results of our deconvolution process in Supplementary Fig. 1 show that the signal on $m/z = 77$ is underestimated, which may indicate missed out fragments of heteroatom-bearing molecules. Indeed, several heteroatom-bearing molecules with a benzene-core can be identified, for instance, benzoic acid ($C_7H_6O_2$). Benzoic acid has two characteristic peaks (M on $m/z = 122$ and M-OH on $m/z = 105$) and both are clearly present in our data. The third-most intense peak of the benzoic acid reference spectrum is due to the phenyl cation. Just like benzoic acid, also benzylamine ($C_7H_9N$) has a major fragment on $m/z = 77$, combined, however, with M on $m/z = 107$, M-H on $m/z = 106$, and M-NH$_2$ on $m/z = 91$. As all of these signals are clearly present in the ensemble as certain, we consider the identification of benzylamine in the ensemble as certain. Very likely is the presence of five-membered heterocycles in the form of furane, pyrrole, and thiophene or derivatives of these compounds.

However, there are also heteroatom-based functionalities which do not produce strong molecular ion signals such as some alcohols, ethers, or chain-structured olefins. Especially in a complex mixture like a cometary coma it is very difficult to identify such species. This is a clear bias to our analysis and has to be kept in mind. Vice versa, and as already mentioned above, yet unidentified representatives of such molecules may be an explanation for certain portions of unaccounted intensity. Larger alkanols, such as 1-nonanol, typically lead to fragments of the structure R-CH$_2$, where R is some olefinic residue, while the alcohol group is being lost. The presence of alkanols may explain unaccounted intensity around $m/z = 70$, 84, 98 in the pure hydrocarbon data set.

To conclude: Further work, including extensive, additional calibration of the DFMS flight spare at the University of Bern is necessary to support the identification of further molecules and to increase the confidence level of certain candidates proposed here. The data subset of the different groups of heteroatom-bearing species still has a great potential to help extending the census of cometary molecules.

### Data availability
The datasets evaluated and interpreted in the framework of the presented study, together with a user manual for data analysis, are available via ESA's Planetary Science Archive (PSA; https://www.cosmos.esa.int/web/psa/rosetta) or NASA's Planetary Data System (PDS; https://pdssbn.astro.umd.edu/data_sb/missions/rosetta/index.shtml).

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

## Acknowledgements

ROSINA would not have produced such outstanding results without the work of the many engineers, technicians, and scientists involved in the mission, in the Rosetta spacecraft team, and in the ROSINA instrument team over the past 20 years. Their contributions are gratefully acknowledged. Rosetta is an ESA mission with contributions from its member states and NASA. We acknowledge herewith the work of the whole ESA Rosetta team. Work at the University of Bern was funded by the State of Bern, the Swiss National Science Foundation (SNSF, 200020_182418 & 200020_207312), the Swiss State Secretariat for Education, Research and Innovation (SERI) under contract number 16.0008- 2, and the European Space Agency's PRODEX programme. Work at BIRA-IASB was supported by the Belgian Science Policy Office via PRODEX/ROSINA PEA 90020. S.F.W. acknowledges the financial support of the SNSF Eccellenza Professorial Fellowship PCEFP2_181150. N.H. especially thanks B. A. McGuire for kindly sharing the information contained in his updated census of ISM molecules[60] as well as for his willingness to discuss about apparent differences and similarities between this census and 67P's organic budget.

## Author contributions

N.H. evaluated the data and wrote the manuscript. K.A. was principal investigator of the ROSINA instrument, reduced the data presented in this article, and supported data analysis. M.C., J.D.K., and S.A.F. contributed hardware to the instrument. M.R. operated and calibrated the instrument. S.F.W. especially contributed to the comparison of the presented data to the Interstellar Medium. All authors read and commented on the paper.

## Competing interests

The authors declare no competing interests.
