## [Peer Review File · Nature Communications]

REVIEWER COMMENTS

Reviewer #1 (Remarks to the Author):

I carefully read the manuscript entitled "Evidence for a continuum of cometary organics between volatile and refractory" and do not recommend it for publication in Nature communications unless substantial modifications are implemented and stronger conclusions can be drawn.

While the data analysis and the identification of several linear, cyclic and aromatic organics provide new and interesting information to the cosmochemistry field and should be published, I doubt that the results presented here will be of interest to the broad scientific community. The paper's main point, according to the title, is that cometary organics detected in the coma could originate from an ensemble of single molecules of various volatilities, as opposed to the previous hypothesis that high mass organics would come from the fragmentation of a polymeric substance. The paper did not, in my opinion, illustrate enough the impact of such discovery to warrant publication in a very high impact journal such as Nature communications. For example, the paper mentions that the origin of these cometary organics could either be from top-down or bottom-up chemistry converting and transferring organic matter from the refractory phase to volatiles. Settling between the two options would provide valuable insights on the origin of cometary organics and support or refute a potential inheritance from the presolar nebulae. The results presented here however, do not allow to disentangle between the two scenarios, as rightfully mentioned by the authors. If modeling or experimental work would support one of the two hypothesis, the paper would have a greater impact and likely be of interest to the broader scientific community.

Below are some minor comments that could be helpful for resubmission here or to another journal :

-The paper's main message, according to the title is that sublimating molecules detected by Rosina do not come from POM (or other polymers) but from single molecules instead. The paper however, does not show how sublimating and perhaps decomposed POM (or other polymers) would yield a significantly different mass spectrum. Showing a figure or at least providing a quantification a fit goodness for the ensemble of single molecules versus the polymer signal would strengthen this important point.

- Briefly mention the various or potential origins/age of Saturn's ring rain organic material to put the comet/ring comparison into context. The elemental similarities are interesting though it is not clear from the paper what can be from them for the various ring origin out there (initial planetary disc, shattered moon, capture of meteorites...)

-Figure 2 is quite obscure for a reader not familiar with the specific data analysis technique employed in the paper. It could, in my opinion, be removed from the "main text" since the analysis results are shown in table 2.

-The comparison of the organic content at 67P with the ISM based on the HDI in figure 3 and in the accompanying text is very tenuous. The abstract mentions that the sublimated C-bearing species identified here overlap with ISM molecules, which "may hint at a shared prestellar origin". The figure, however, shows a spread of HID and few species overlap. The text rightfully indicates that millimeter/sub-millimeter and mass spectrometric techniques are sensitive to different type of species, which can explain the spread, but the results show do not support a common origin hypothesis. I suggest not mentioning a potential ISM-cometary link in the abstract.

-L344 "In a cometary environment, top-down processes, like the degradation of refractory organics in the presence of hydrogen sufficient to increase H/C appropriately, seem rather

unlikely, especially close to the comet, as for the case of Rosetta." This statement needs to be developed and supported. Cometary H₂O and macromolecules subject to solar radiation or other particles may (or may not) yield small amounts of organics with a high H/C ratio. I don't think such reactions are "rather unlikely", a priori.

Reviewer #2 (Remarks to the Author):

The paper "Evidence for a continuum of cometary organics between volatile and refractory" is of high interest and addresses the subject of the identification of organic compounds and their classifications as detected in-situ in the coma during perihelion passage of the comet 67P/Churyumov-Gerasimenko. Some compounds are uniquely identified in a cometary coma for the first time, others are identified with a high level of confidence. The organic compounds are classified and quantified as chain-based, cyclic, and aromatic species. The composition is compared to the refractory organic cometary composition as well as the organics identified in the Saturn rings. Polyoxymethylene, a polymer proposed as a source for the organic peaks observed in the coma of Comet 1P/Halley, is ruled out by this new results.

The work is original and provides significant results which are state of art in the field of the composition of cometary matter. Organic molecules can only be uniquely identified with a high resolution mass spectrometer to reduce interferences.

The work describes in detail in the methodology section how the results are derived.

Some minor points, which might need to be addressed, are :

- The signal or peak intensities are labelled in arbitrary units while in the paper, the statistical error is quantified and it seems to be lower than the systematic errors. It would improve the paper if the real counts are applied in the figures to support the statistical error statements. Further, the authors do not explain in detail why they are not making any effort to extend the analysis period to improve a) significantly the statistical error and b) further, since the identification is based on the sum of the fragmentation patterns of molecules by electron impact, variation of the molecular parent intensities with time should significantly improve the identification. If for operational or physical reasons this was not possible, it should be explained why only a very brief period is the basis for the analysis applied in the paper and why no improvement is expected with an extended database.
- The theory based on polyoxymethylene or POM as the parent molecule for the observations in the coma of 1P/Halley was modelled with an extended source, as not only the nucleus, but also the dust particles being an additional source for the observed patterns in the mass spectra and explaining the signal decay with nucleus distance. In the paper, only one distance off the nucleus is studied and the extended source is not discussed. The observed zoo of organic molecules should show a different picture as extended source than the previous POM models. Further, already in 1989, Mitchell et al. put forward the idea that the pattern observed in the coma of 1P/Halley were not necessarily due to a single polymer (Mitchell, D. L., Lin, R. P., Anderson, K. A., & Carlson, C. (1989). Complex organic ions in the atmosphere of comet Halley. *Advances in Space Research*, 9(2), 35.).
- As the statistical error is 1 sigma and the systematic errors have to be added, the stated ratios of the chemical classes should be linked to a stated overall error. Only based on that, one might conclude that the composition of the organic refractory and volatile matter is similar. However, even if the chemical sum-formula are significantly identical, the actual molecular composition might be very different. The cometary refractory composition was compared to the IOM composition of selected meteorites in the past and the similarity or dis-similarity of the observed chemical composition with the known IOM of various meteorite classes is not addressed. This is in so far an interesting point, as the differences in insoluble and soluble organic composition

(IOM and SOM) of meteoritic matter is well known and in the context of the drawn conclusion that refractory and volatile matter is of the same class worth to be addressed in the discussion.

ANSWERS TO REVIEWER COMMENTS

We thank both reviewers for the careful reading and the valuable comments which guided our revisions and helped to substantially improve coherence of our manuscript. All changes to the manuscript are highlighted in red as are our point-by-point responses to your comments below. Please note that we also revised the language, which resulted in additional minor changes to the text without actually changing content or meaning.

Reviewer #1 (Remarks to the Author):

I carefully read the manuscript entitled “Evidence for a continuum of cometary organics between volatile and refractory” and do not recommend it for publication in Nature communications unless substantial modifications are implemented and stronger conclusions can be drawn.

While the data analysis and the identification of several linear, cyclic and aromatic organics provide new and interesting information to the cosmochemistry field and should be published, I doubt that the results presented here will be of interest to the broad scientific community. The paper’s main point, according to the title, is that cometary organics detected in the coma could originate from an ensemble of single molecules of various volatilities, as opposed to the previous hypothesis that high mass organics would come from the fragmentation of a polymeric substance. The paper did not, in my opinion, illustrate enough the impact of such discovery to warrant publication in a very high impact journal such as Nature communications. For example, the paper mentions that the origin of these cometary organics could either be from top-down or bottom-up chemistry converting and transferring organic matter from the refractory phase to volatiles. Settling between the two options would provide valuable insights on the origin of cometary organics and support or refute a potential inheritance from the presolar nebulae. The results presented here however, do not allow to disentangle between the two scenarii, as rightfully mentioned by the authors. If modeling or experimental work would support one of the two hypothesis, the paper would have a greater impact and likely be of interest to the broader scientific community.

We understood that it was misleading to employ the terminology of ‘bottom-up’ and ‘top-down’ formation in the context of the observed cometary molecules and we wish to state clearly that it is beyond the scope of our work to make statements about possible material processing in the ISM. However, we are of the opinion that the observed molecules are likely inherited from stages of the Solar System history prior to comet formation and we supported our line of argumentation with a solid body of literature relevant for the case of comet 67P, including a new work of our own group led by our D. Müller [Müller et al. A&A 2022, forthcoming], see further details below.

In response to this comment, we completely revised our manuscript. The new version, with a changed title, 1) identifies, characterises, and inter-compares an ensemble of cometary complex organics (including several previously unidentified molecules), 2) excludes the relevance of cometary polymeric matter, and 3) presents a plausible scenario of origin of the observed molecules. To strengthen these three points, we included additional evidence supporting the absence of two high-ranking candidates of cometary polymers, namely, POMs and HMT (ad 2). Derivation of further descriptive parameters to characterize organics (we newly introduced the $sp^2:sp^3$ and the $CH_2:CH_3$ ratios) strengthens the comparison/contrast of our ensemble with other reservoirs of organics in the Solar System and beyond (ad 1). We argue that the compositional and structural properties of 67P’s volatile organics are compatible with a scenario of pre-solar origin and subsequent processing in different environments. Such a scenario could explain the observation that heteroatom-abundances are constant while the degree of hydrogenation considerably varies between the different reservoirs of organics (ad 3).

We are convinced that these findings are relevant to a broad community of researchers well beyond the field of astro-/cosmochemistry and that Nature Communications is thus a suitable journal for the publication of our work.

Below are some minor comments that could be helpful for resubmission here or to another journal:

-The paper's main message, according to the title is that sublimating molecules detected by Rosina do not come from POM (or other polymers) but from single molecules instead. The paper however, does not show how sublimating and perhaps decomposed POM (or other polymers) would yield a significantly different mass spectrum. Showing a figure or at least providing a quantification a fit goodness for the ensemble of single molecules versus the polymer signal would strengthen this important point.

We agree with this comment, saying that we did not detail sufficiently our arguments against the relevance of polymeric matter to the observed chemical complexity in 67P's dusty coma at perihelion. In response, we combined in the revised manuscript the Occam's razor conform explanation of the DFMS data with specific evidence for the absence of the two most relevant polymeric candidates, namely, POMs and HMT, see new Fig. 2. Furthermore, we included additional relevant literature both in the last paragraph of the 'Introduction' section as well as in a new paragraph in the 'Results' section ('Absence of polymers', lines 197-234), highlighting that calibration experiments have been performed on the laboratory twin instruments previously [Altwegg et al. 2017] and explaining why the calibrated patterns are incompatible with the DFMS data. We would like to stress, however, that the exclusion of the relevance of cometary polymeric matter is an important finding of our work (and a historic anchor-point of our work) but it is clearly not the only one, cf. response above.

-Briefly mention the various or potential origins/age of Saturn's ring rain organic material to put the comet/ring comparison into context. The elemental similarities are interesting though it is not clear from the paper what can be from them for the various ring origin out there (initial planetary disc, shattered moon, capture of meteorites...)

We expanded the respective paragraph, addressing briefly the different scenarios of origin of the Saturnian ring system while highlighting the potential importance of the study of the organic material in the rings (cf. lines 397 ff.).

-Figure 2 is quite obscure for a reader not familiar with the specific data analysis technique employed in the paper. It could, in my opinion, be removed from the "main text" since the analysis results are shown in table 2.

We agree that (original) Fig. 2 is complicated for the non-expert reader and we therefore moved it to the Supplementary material (now Supplementary Fig. 1).

-The comparison of the organic content at 67P with the ISM based on the HDI in figure 3 and in the accompanying text is very tenuous. The abstract mentions that the sublimated C-bearing species identified here overlap with ISM molecules, which "may hint at a shared prestellar origin". The figure, however, shows a spread of HID and few species overlap. The text rightfully indicates that millimeter/sub-millimeter and mass spectrometric techniques are sensitive to different type of species, which can explain the spread, but the results show do not support a common origin hypothesis. I suggest not mentioning a potential ISM-cometary link in the abstract.

We reworked and expanded our manuscript substantially as a consequence of this comment, holding, however, to our original line of argument and to our final sentence of the abstract (lines 32 ff.). Details have been outlined above. We find the inheritance-scenario to be compatible with our data while arguing that formation of the detected molecules subsequent to the formation of the comet itself, e.g., from decomposing macromolecules, is unlikely. This view, supported by a solid body of literature,

is presented in the new tailing paragraphs of the 'Results' section (lines 407 ff.), which now also include some of the content of the original 'Discussion' section (the 'Discussion' section was removed). The paragraph on the HDI, addressed in this comment, remained unchanged. We believe that, given the potential inheritance scenario of the cometary organics, some of the new organic molecules we successfully identified in 67P's neutral coma could be interesting target species for future observations of dusty environments in the ISM - despite the indicated methodological differences.

-L344 "In a cometary environment, top-down processes, like the degradation of refractory organics in the presence of hydrogen sufficient to increase H/C appropriately, seem rather unlikely, especially close to the comet, as for the case of Rosetta." This statement needs to be developed and supported. Cometary H₂O and macromolecules subject to solar radiation or other particles may (or may not) yield small amounts of organics with a high H/C ratio. I don't think such reactions are "rather unlikely", a priori.

We have several solid arguments supporting a scenario where the molecules detected by ROSINA/DFMS in the inner coma of comet 67P have been inherited from times of the Solar System history preceding comet formation and not formed at the comet, e.g., from decomposing refractory organics. Responding to this comment, we listed these arguments, in great detail and well-supported by published literature, in the text added to conclude the 'Results' section of the manuscript (cf. lines 437 ff.).

Reviewer #2 (Remarks to the Author):

The paper "Evidence for a continuum of cometary organics between volatile and refractory" is of high interest and addresses the subject of the identification of organic compounds and their classifications as detected in-situ in the coma during perihelion passage of the comet 67P/Churyumov-Gerasimenko. Some compounds are uniquely identified in a cometary coma for the first time, others are identified with a high level of confidence. The organic compounds are classified and quantified as chain-based, cyclic, and aromatic species. The composition is compared to the refractory organic cometary composition as well as the organics identified in the Saturn rings. Polyoxymethylene, a polymer proposed as a source for the organic peaks observed in the coma of Comet 1P/Halley, is ruled out by this new results.

The work is original and provides significant results which are state of art in the field of the composition of cometary matter. Organic molecules can only be uniquely identified with a high resolution mass spectrometer to reduce interferences.

The work describes in detail in the methodology section how the results are derived.

Some minor points, which might need to be addressed, are:

- The signal or peak intensities are labelled in arbitrary units while in the paper, the statistical error is quantified and it seems to be lower than the systematic errors. It would improve the paper if the real counts are applied in the figures to support the statistical error statements. Further, the authors do not explain in detail why they are not making any effort to extend the analysis period to improve a) significantly the statistical error and b) further, since the identification is based on the sum of the fragmentation patterns of molecules by electron impact, variation of the molecular parent intensities with time should significantly improve the identification. If for operational or physical reasons this was

not possible, it should be explained why only a very brief period is the basis for the analysis applied in the paper and why no improvement is expected with an extended database.

We use arbitrary units, because we cannot give a precise number for the ions as this would need calibration of the detector for all species mentioned in this paper. The detector is operated in analogue mode and its efficiency depends mostly on the ion energy (which we account for), but unfortunately also on the exact shape of the molecule/fragment (number of atoms and geometry of the species). The intensity in a.u. is therefore for individual spectra only an approximate number of ions, for the total intensity it is the area under the peak. For the error we do not only account for the approximate number of ions, we also account for a (statistical) fitting error and statistical error due to the individual pixel gain. Uncertainties have already been addressed in detail in the 'Methods' section (subsection 'Instrumentation and data reduction', lines 506 ff.).

Unfortunately, we had only very few times, when we could operate DFMS in a steady state, with the S/C looking Nadir. The measurements were very often interrupted by wheel-offloadings, where, for safety reasons, DFMS was off, or by slewing in all directions for the needs of the other payload. In addition, of course, the comet rotated by 30° per hour underneath the S/C. The S/C often was at terminator, meaning the latitude changed continuously. Especially during perihelion, the S/C was not in a bound orbit, which meant, that also the distance to the comet changed. As detailed in the 'Methods' section (subsection 'Instrumentation and data reduction') already, spacecraft slewing did, also in the short time-span of data acquisition considered in the presented analysis, cause a drift in the data for which we had to correct. In response to this comment, we included a new sentence to address the stacking of data specifically, see lines 522-523: 'Due to the highly variability of the observational conditions as indicated above, stacking of mass spectra does not result in a reduction of uncertainty but rather the opposite is the case.' Note that the only time when our instrument had priority was far from the Sun, when we could stack spectra to detect the noble gas Xe.

- The theory based on polyoxymethylene or POM as the parent molecule for the observations in the coma of 1P/Halley was modelled with an extended source, as not only the nucleus, but also the dust particles being an additional source for the observed patterns in the mass spectra and explaining the signal decay with nucleus distance. In the paper, only one distance off the nucleus is studied and the extended source is not discussed. The observed zoo of organic molecules should show a different picture as extended source than the previous POM models. Further, already in 1989, Mitchell et al. put forward the idea that the pattern observed in the coma of 1P/Halley were not necessarily due to a single polymer (Mitchell, D. L., Lin, R. P., Anderson, K. A., & Carlson, C. (1989). Complex organic ions in the atmosphere of comet Halley. *Advances in Space Research*, 9(2), 35.).

Although we agree with reviewer #2, the investigation of extended sources commonly explained with decomposing polymers or the search alternative explanations for the reported extended source of formaldehyde and carbon monoxide surpasses the scope of this work and the scope of the data presented in it. The manuscript presents and analyses data from one cometocentric distance only. However, in response to this comment, we acknowledge in the introductory part of the manuscript (cf. lines 63 ff.) that decomposing polymers, POMs first, were suggested as a potential explanation for the observed extended density profiles of some cometary species, see Maier et al. (1993). We also mention that Rubin et al. (2011) successfully modelled observed densities by simply allowing for a time-variable cometary activity, which means that no extended source may be needed.

The publication by Mitchell et al. (1989) has evaded our attention and we thank reviewer #2 for their notice. Mitchell et al. (1989) brought forth the idea of independent molecules to explain the PICCA observations that were interpreted to indicate polymeric matter by Huebner (1987). Their results were published two years after those of Kissel and Krueger (1987), who already interpreted PUMA data accordingly, as we have mentioned in the original introductory section. In response to this comment and in order to improve completeness and clarity of the scientific case for the non-expert reader, we rewrote the tailing paragraph of the 'Introduction' section that pertains to the 1P/Halley measurements, cf. lines 56-77. We highlight that only high-resolution mass-spectrometric data, as

recently provided by the DFMS on Rosetta, allow to unambiguously distinguish between the opposing ideas of polymeric matter versus individual complex molecules. This is one major aim (but not the only one) of our work.

- As the statistical error is 1 sigma and the systematic errors have to be added, the stated ratios of the chemical classes should be linked to a stated overall error. Only based on that, one might conclude that the composition of the organic refractory and volatile matter is similar. However, even if the chemical sum-formula are significantly identical, the actual molecular composition might be very different. The cometary refractory composition was compared to the IOM composition of selected meteorites in the past and the similarity or dis-similarity of the observed chemical composition with the known IOM of various meteorite classes is not addressed. This is in so far an interesting point, as the differences in insoluble and solvable organic composition (IOM and SOM) of meteoritic matter is well known and in the context of the drawn conclusion that refractory and volatile matter is of the same class worth to be addressed in the discussion.

In response to this comment, we estimated the uncertainty of the ratio of the different groups of pure hydrocarbon molecules represented in our ensemble (cf. Table 1). The ratio estimates (~6:3:1) have an uncertainty of 10%, which includes statistical as well as systematic uncertainties. However, this estimation does not consider heteroatom-bearing molecules and it may therefore not be the same for the full ensemble. To clarify how exactly we derived this ratio and what the uncertainty on it is, we have rewritten the corresponding text in the 'Results' section. It now reads (lines 166-170): 'Note that aromatic molecules are also cyclic but aromaticity is prioritized and no double-assignments are made. Based on the relative abundance estimates corresponding to the fragment sum of the individual pure hydrocarbon molecules in Table 1 an approximate ratio of 6:3:1 is obtained. This ratio estimate has a 1σ uncertainty of 10%, which includes statistical as well as systematic uncertainties.'

It is clear that the average sum formula of a specific organic reservoir alone does not allow conclusions on the molecules present. This is exactly what makes our case so special in the framework of analysis of extraterrestrial reservoirs of organics: We can also pin down some individual molecules of our ensemble thanks to our Occam's razor compatible deconvolution approach. We thank reviewer #2 for indicating to us the body of literature related to meteoritic SOM, which we think is a very important additional reservoir of organic molecules for inter-comparison. In response to this comment, we included SOM with an additional line in Table 2. Instead of Alexander et al. (2007), we now refer to Alexander et al. (2017) for IOM and to Schmitt-Kopplin et al. (2010) for SOM. In the corresponding main text ('Average composition'), we also completely rewrote the short paragraph on organics in meteorites (lines 374-383), highlighting that average SOM (from Murchinson) actually shows the closest agreement (even regarding the H/C ratio) with the average sum formula derived for 67P's volatile complex organics.

REVIEWERS' COMMENTS

Reviewer #1 (Remarks to the Author):

I compliment the authors for all the substantial changes they made to the manuscript.

The case for a non polymeric origin is now well argued.

The sp²:sp³, CH₂:CH₃ ratios will be of high interest to both the astro and the meteorite/sample return communities.

I especially appreciate the additional elemental comparison to Murchison's SOM. The potential link between the two is a great find that will make an impact.

The discussion at the end of the paper summarizing the evidence for an inheritance scenario of organics from the dark clouds to the SS is nicely bringing the paper's results into context.

I'm still, however, having issues with the paragraph about the HDI comparison for 67P vs ISM, p237.

As a reader, I currently do not understand why looking at these HDI versus number of atoms is important and why you are attempting to compare the HDI versus number of atoms for the ISM and the 67P.

I get that you have plenty of evidence for an ISM inheritance of the organics from complementary observations/calculations/experiments (side note, I'm convinced by the inheritance scenario too), but what is the reader supposed to gather from the comparison you make in figure 3 of the HDI for 67P and the ISM versus number of atoms?

To help us get the point of that comparison (and if there is a point), I suggest you briefly describe figure 3 on line 246 (e.g. do the two datasets overlap substantially? Do the two datasets display a similar HDI spread, despite the different number of atoms that can be probed in the ISM and for 67P? Is there a trend for HDI vs number of atoms?). After that, I suggest you specify what you're learning from the comparison (does the HDI spread and the HDI versus number of atoms tell you something about the the 67P organics origin? Do you expect a difference HDI spread for 67P for an ISM inheritance and an in-situ formation? What does the trend of HDI versus number of atoms tell you?). Add these information also in the caption of figure 3.

You write on line 282 "We cannot conclude from this comparison whether or not species represented in both the cometary and the interstellar environment have formed under similar astrophysical conditions. However, both techniques and perspectives substantially add to a comprehensive picture of the composition and compositional evolution of cosmomaterials."

If there is really no information to be gathered from the HDI comparison at present, I suggest not presenting this HDI plot and paragraph in the results section but moving them in the discussion section, at the end of the paper (after the paragraph where you give us the evidence for the inheritance scenario), and framing this HDI discussion around what you brought up in the rebuttal "We believe that, given the potential inheritance scenario of the cometary organics, some of the new organic molecules we successfully identified in 67P's neutral coma could be interesting target species for future observations of dusty environments in the ISM - despite the indicated methodological differences." You could perhaps recommend future ISM molecular searches to focus on specific molecule pairs with widely different HDI to help in our quest for validating/discarding the organics inheritance/in-situ formation scenarii (it's just a suggestion).

That would make the narrative more logical since, as currently written, we are expecting a result from the HDI ISM 67P comparison and you are not giving us any.

Reviewer #2 (Remarks to the Author):

The paper now titled "Identification and characterization of a new ensemble of cometary organic molecules" is of high interest and addresses the subject of the identification of organic compounds and their classifications as detected in-situ in the coma during perihelion passage of the comet 67P/Churyumov-Gerasimenko. Some compounds are uniquely identified in a cometary coma for the first time, others are identified with a high level of confidence. The organic compounds are classified and quantified as chain-based, cyclic, and aromatic species. The composition is compared to the refractory organic cometary composition as well as the organics identified in the Saturn rings. Polyoxymethylene as well as

hexamethylenetetramine, both polymers proposed previously as sources for the organic peaks observed in the coma of Comet 1P/Halley, are ruled out by this new results.

The work is original and provides most significant results which are state of art in the field of the composition of cometary matter. Organic molecules can only be uniquely identified with a high resolution mass spectrometer to reduce interferences, in-situ observed close to the nucleus and during the perihelion passage of the comet.

The minor points raised in the previous review have been addressed and well clarified. In the case of the extended source, the line of argument based on the high resolution mass spectra is convincing as outlined in the revised version.

ANSWERS TO REVIEWER COMMENTS

We thank both reviewers for their careful reading and the valuable and constructive comments and suggestions of reviewer #1 which guided our final revisions and helped to improve the manuscript's narrative. All changes to the manuscript are highlighted in red as is our detailed response to reviewer #1 below.

REVIEWERS' COMMENTS

Reviewer #1 (Remarks to the Author):

I compliment the authors for all the substantial changes they made to the manuscript. The case for a non polymeric origin is now well argued. The sp²:sp³, CH₂:CH₃ ratios will be of high interest to both the astro and the meteorite/sample return communities. I especially appreciate the additional elemental comparison to Murchison's SOM. The potential link between the two is a great find that will make an impact. The discussion at the end of the paper summarizing the evidence for an inheritance scenario of organics from the dark clouds to the SS is nicely bringing the paper's results into context.

I'm still, however, having issues with the paragraph about the HDI comparison for 67P vs ISM, p237. As a reader, I currently do not understand why looking at these HDI versus number of atoms is important and why you are attempting to compare the HDI versus number of atoms for the ISM and the 67P.

I get that you have plenty of evidence for an ISM inheritance of the organics from complementary observations/calculations/experiments (side note, I'm convinced by the inheritance scenario too), but what is the reader supposed to gather from the comparison you make in figure 3 of the HDI for 67P and the ISM versus number of atoms?

To help us get the point of that comparison (and if there is a point), I suggest you briefly describe figure 3 on line 246 (e.g. do the two datasets overlap substantially? Do the two datasets display a similar HDI spread, despite the different number of atoms that can be probed in the ISM and for 67P? Is there a trend for HDI vs number of atoms?). After that, I suggest you specify what you're learning from the comparison (does the HDI spread and the HDI versus number of atoms tell you something about the the 67P organics origin? Do you expect a difference HDI spread for 67P for an ISM inheritance and an in-situ formation? What does the trend of HDI versus number of atoms tell you?). Add these information also in the caption of figure 3.

You write on line 282 "We cannot conclude from this comparison whether or not species represented in both the cometary and the interstellar environment have formed under similar astrophysical conditions. However, both techniques and perspectives substantially add to a comprehensive picture of the composition and compositional evolution of cosmomaterials."

Given the likely scenario of ISM inheritance of cometary species, Figure 3 attempts a visual comparison of two censuses (ISM and cometary), allowing the reader to grasp at one glimpse similarities and differences. Both the observed species' size (number of atoms) and their degree of hydrogen deficiency (HDI) and, hence, their chemical structure are represented. This figure and the corresponding discussion, however, was not contributing any result to the manuscript as noticed by the reviewer #1. In addition to methodological biases and limitations, issues like decreasing instrument sensitivity with increasing m/z or decreasing volatility with increasing mass of the analyte further complicate even the interpretation of the cometary mass-spectrometry-based data set. We are therefore hesitant to see trends in our data or interpret them based on statistical indicators and decided to follow the reviewer #1's suggestion below.

If there is really no information to be gathered from the HDI comparison at present, I suggest not presenting this HDI plot and paragraph in the results section but moving them in the discussion section, at the end of the paper (after the paragraph where you give us the evidence for the inheritance scenario), and framing this HDI discussion around what you brought up in the rebuttal “We believe that, given the potential inheritance scenario of the cometary organics, some of the new organic molecules we successfully identified in 67P’s neutral coma could be interesting target species for future observations of dusty environments in the ISM - despite the indicated methodological differences.” You could perhaps recommend future ISM molecular searches to focus on specific molecule pairs with widely different HDI to help in our quest for validating/discarding the organics inheritance/in-situ formation scenarii (it’s just a suggestion).

That would make the narrative more logical since, as currently written, we are expecting a result from the HDI ISM 67P comparison and you are not giving us any.

We agree with reviewer #1’s suggestion to move the content of the paragraph under debate (including Fig. 3) to the end of the Results and discussion section of the main text and present it rather as an outlook than as a result. Tying the ISM-comet comparison (lines 379-450) to the likely inheritance scenario, which was discussed just previously (lines 341-379), the manuscript’s narrative becomes more logical and easier to follow for the reader.

Starting on line 379 ff. and connecting to the previously tailing sentences, we compare and contrast our cometary census to the ISM census of organics based on the species’ HDIs vs. no. of atoms (Fig. 3). Fig. 3 and the data it visualises are subsequently discussed with respect to methodological biases and limitations of the methods of rotational spectroscopy and mass spectrometry (now presented on lines 390-408 in a more detailed and restructured way), suggesting, that both perspectives can valuably add to a comprehensive picture of the composition and compositional evolution of cosmomaterials. Subsequently, we discuss differences evident from Figure 3 (line 409 ff.). While we removed a few mentions of / references to specific molecules identified in TMC-1, we added (also to Fig. 3) a recently published search for a series of heterocyclic species *ibidem* (Baranum et al. 2022), which, however, yielded upper limits only. Referencing the results of Barnum et al. (2022), we suggest that analyses of in-situ high-resolution mass-spectrometric cometary data (e.g., ROSINA/DFMS from ESA’s Rosetta mission) can be guided by such studies. In the future, we plan to make further efforts to extend the cometary census for heteroatom-bearing (incl. heterocyclic) species, which is a time-consuming project that goes beyond the scope of this work. And vice versa, we clearly would appreciate future observational efforts tying to our work and other comet studies, e.g., while following-up on the results of Barnum et al. (2022).

With this outlook, we hope to smoothly complete the narrative of our main text. We also value the very specific suggestion of reviewer #1 to ‘recommend future ISM molecular searches to focus on specific molecule pairs with widely different HDI to help in our quest for validating/discarding the organics inheritance/in-situ formation scenarii’. Although we consider it an interesting idea, we believe that without accompanying extensive modelling/laboratory work, it would be tricky to use, e.g., the HDI spread to either support or refute the inheritance scenario.

Reviewer #2 (Remarks to the Author):

The paper now titled “Identification and characterization of a new ensemble of cometary organic molecules” is of high interest and addresses the subject of the identification of organic compounds and their classifications as detected in-situ in the coma during perihelion passage of the comet 67P/Churyumov-Gerasimenko. Some compounds are uniquely identified in a cometary coma for the first time, others are identified with a high level of confidence. The organic compounds are classified and quantified as chain-based, cyclic, and aromatic species. The composition is compared to the refractory organic cometary composition as well as the organics identified in the Saturn rings.

Polyoxymethylene as well as hexamethylenetetramine, both polymers proposed previously as sources for the organic peaks observed in the coma of Comet 1P/Halley, are ruled out by this new results.

The work is original and provides most significant results which are state of art in the field of the composition of cometary matter. Organic molecules can only be uniquely identified with a high resolution mass spectrometer to reduce interferences, in-situ observed close to the nucleus and during the perihelion passage of the comet.

The minor points raised in the previous review have been addressed and well clarified. In the case of the extended source, the line of argument based on the high resolution mass spectra is convincing as outlined in the revised version.